# Mental Health Conditions According to Stress and Sleep Disorders

**DOI:** 10.3390/ijerph19137957

**Published:** 2022-06-29

**Authors:** Ray M. Merrill

**Affiliations:** Department of Public Health, Brigham Young University, Provo, UT 84602, USA; ray_merrill@byu.edu; Tel.: +1-801-422-9788

**Keywords:** hypersomnia, insomnia, mental health disorders, modifier, sleep apnea, stress

## Abstract

The purpose of this study was to compare associations between stress and sleep disorders (insomnia, hypersomnia, and sleep apnea), identify potential modifying effects, and compare associations between stress and types of sleep disorders with selected mental health conditions. Analyses were based on 21,027 employees aged 18–64 years in 2020 who were insured by the Deseret Mutual Benefit Administrators (DMBA). The risk of stress (2.3%) was significantly greater in women, singles, and those with dependent children. The risk of a sleep disorder was 12.1% (2.1% for insomnia, 1.0% for hypersomnia, and 10.1% for sleep apnea). The risk of stress was significantly greater for those with a sleep disorder (136% overall, 179% for insomnia, and 102% for sleep apnea after adjusting for age, sex, marital status, dependent children, and sleep disorders). The risk of stress among those with sleep apnea was significantly greater for singles than for married individuals. Approximately 9.5% had anxiety, 8.5% had depression, 2.0% had ADHD, 0.6% had bipolar disorder, 0.4% had OCD, and 0.1% had schizophrenia. Each of these mental health conditions was significantly positively associated with stress and sleep disorders. Bipolar disorder and schizophrenia were more strongly associated with stress and sleep disorders than were the other mental health conditions. Insomnia was more strongly associated with anxiety, bipolar disorder, OCD, and schizophrenia than was sleep apnea.

## 1. Introduction

Stress can lead to an immune response in the body and consequential adverse health outcomes. Psychological distress signals to the body a deviation from homeostasis, triggering a reaction from the immune system, which often manifests itself in the form of inflammation [1]. The longer that stress persists, the less the body can manage the associated prolonged inflammation, which increases the risk of susceptibility to a wide range of illnesses [2,3]. Stress over time can lead to immune diseases such as psoriasis and rheumatoid arthritis [4]. Stress and inflammation are contributing factors to cardiovascular diseases, including coronary heart disease and atherosclerosis [5,6]. Inflammation also contributes to several respiratory diseases, such as chronic obstructive pulmonary disease (COPD), asthma, and pulmonary fibrosis [7].

Stress is also associated with sleep disorders [8,9,10,11]. Stress and sleep disorders have a bidirectional relationship that affects the central nervous system and metabolism [12,13]. High stress hormone levels are associated with decreased sleep duration, with both stress and sleep problems being associated with obesity and metabolic syndrome [14]. Metabolic syndrome is a cluster of conditions (high blood pressure, blood sugar, cholesterol or triglycerides, and excessive body fat around the waist), which increase the risk of type 2 diabetes, stroke, and heart disease. 

Common sleep disorders include insomnia (difficulty falling and/or staying asleep), hypersomnia (excessive daytime sleepiness), and sleep apnea (airflow is limited while sleeping, causing low oxygen saturation and disrupted sleep). Many important physiological changes take place in all body systems and organs during sleep, but with insufficient sleep, physical and mental health problems ensue [15]. For example, symptoms associated with insomnia, such as prolonged sleep latency, problems maintaining sleep, and early morning awakening, are associated with an increased risk of illness, cognitive impairment, lower productivity, and mental health problems [16,17,18].

The association between stress and sleep disorders is complex and often bidirectional, with both direct and indirect influences. Stress can cause insomnia and be the result of it, thereby explaining the positive association between stress and insomnia that is often observed [19]. Hypersomnia may be caused by another sleep disorder (e.g., sleep apnea), dysfunction of the autonomic nervous system, a physical problem (e.g., tumor, head trauma, or injury to the central nervous system), a medical condition (multiple sclerosis, depression, or obesity), or drug or alcohol abuse [20]. It has been suggested that hypersomnia can cause anxiety [21], but anxiety may also indirectly contribute to hypersomnia by influencing other sleep disorders or drug or alcohol abuse. Obstructive sleep apnea can contribute to stress by preventing restful sleep, resulting in an elevation of the stress hormone cortisol [22]. Further, stress can contribute to unhealthy habits (e.g., smoking, alcohol abuse, consuming excess caffeine, overeating, and not exercising), which can elevate the risk of sleep apnea [23,24]. In general, each of these sleep disorders has been shown to be a source of stress, creating a cyclical effect that continually harms physical and psychological health [25,26,27].

The current study was conducted to better understand the association between stress and sleep disorders (insomnia, hypersomnia, and sleep apnea), identify potential modifying effects, and determine the relative influence of stress alone, sleep disorders alone, and stress and sleep disorders combined on selected types of mental health illness. 

## 2. Materials and Methods

### 2.1. Study Population

This study was based on demographic and insurance claims data for contract holders employed by the Church of Jesus Christ of Latter-day Saints. Data were obtained from the Deseret Mutual Benefit Administrators (DMBA), a large insurance company that provides health benefits and retirement income to employees and their families. Contract holders in this study resided in Utah (74.2%), Idaho (9.2%), Pacific states (9.1%), and other American states (7.5%). They work in the Church education system, seminaries, and institutes (36.2%); as manual laborers (33.0%); in other companies (10.6); and in other capacities (20.2%). As contract holders retire from work at age 65 years or older and become eligible for Medicare, they tend to be dropped from DMBA.

Employee eligibility data were linked to automated medical claims data. The linked database used in the study was then de-identified according to Health Insurance Portability and Accountability Act (HIPAA) guidelines. The authors’ Institutional Review Board reviewed the study and deemed it exempt from most of the requirements of the Federal Policy for the Protection of Human Subjects.

### 2.2. Data Collection

Analyses are based on 21,027 employed contract holders aged 18–64 years in 2020. International Classification of Diseases, Tenth Revision, Clinical Modification (ICD-10-CM) codes [28] were used to classify the sleep disorders and mental health conditions considered. Licensed health-care providers submitted medical claims to the insurance company. Sleep disorders (G47) were specified as insomnia (G470), hypersomnia (G471), and sleep apnea (G473). Stress (F43) consists of a reaction to severe stress and adjustment disorder. It is applicable to an acute crisis reaction, an acute reaction to stress, a combat and operational stress reaction, combat fatigue, a crisis state, and psychic shock. Specific mental health disorders considered were anxiety (F40, F1), depression (F32, F33), attention deficit hyperactivity disorder (ADHD) (F90), bipolar disorder (F31), obsessive-compulsive disorder (OCD) (F42), and schizophrenia (F20–F29). Other variables considered in the study were age (18–29, 30–39, 49–49, or 50–64), sex, marital status (married or single), and whether the contract holder had one or more dependent children (yes or no). A dependent child was defined as an individual residing in the contract holder’s household who can legally be claimed as a dependent on federal income tax.

### 2.3. Statistical Techniques

Descriptive measures used to characterize the data included counts, means, standard deviations, and percentages. Log-binomial regression models were used to estimate risk ratios adjusted for age, sex, marital status, and dependent children. These variables were available in the DMBA database and considered potential confounders because of their associations with stress and sleep disorders. Risk ratios measuring the association between stress and each specific type of sleep disorder were further adjusted for the other types of sleep disorders. Interaction terms were assessed in the models. The risk of other mental health conditions according to stress and sleep disorders were also estimated in the models. Two-sided tests of significance were used based on the 0.05 level. Statistical analyses were derived from Statistical Analysis System (SAS) software, version 9.1 (SAS Institute Inc., Cary, NC, USA, 2003).

## 3. Results

The average age was 46.42 (SD = 11.50), with 9.9% aged 18–29, 21.5% aged 30–39, 26.8% aged 40–49, and 41.8% aged 50–64; 68.6% men; 78.7% married; and 66.6% with dependent children. The risk of stress (2.3% overall) was significantly greater in women, singles (only for the model involving stress with/without mental illness), and those with dependent children (Table 1). Interaction terms involving sex were evaluated for both models. In the first model, where the outcome variable was stress with/without mental illness, the higher risk of stress among women was significantly greater for singles (Risk Ratio = 3.25, 95% CI 2.05–5.16) than for married individuals (Risk Ratio = 1.94, 95% CI 1.54–2.44). The higher risk of stress among women was also greater for those with no dependent children (Risk Ratio = 2.70, 95% CI 1.88–3.87) versus those with dependent children (Risk Ratio = 1.97, 95% CI 1.55–2.52). In the second model, where the outcome variable was stress without mental illness, the higher risk of stress among women was significantly greater for those with no dependent children (Risk Ratio = 3.58, 95% CI 1.98–6.47) versus those with dependent children (Risk Ratio = 2.24, 95% CI 1.05–4.81). Other interactions involving sex were not significant. 

The risk of a sleep disorder was 12.1%. It was 2.1% for insomnia, 1.0% for hypersomnia, and 10.1% for sleep apnea. Of those with insomnia, hypersomnia, or sleep apnea, 74.6% had sleep apnea only, 13.2% had insomnia only, 6.2% had hypersomnia and sleep apnea, 3.7% had insomnia and sleep apnea, 1.3% had hypersomnia only, 0.8% had insomnia, hypersomnia, and sleep apnea, and 0.2% had insomnia and hypersomnia. These risks of insomnia, hypersomnia, and sleep apnea significantly increased with age, more so for sleep apnea (Table 2). Women had a significantly higher risk of insomnia but a lower risk of hypersomnia or sleep apnea. Married individuals had a significantly greater risk of sleep apnea than singles, whereas those with dependent children had a significantly lower risk of sleep apnea than those without dependent children.

Interaction terms involving sex were evaluated for each model. In the model where sleep disorder was the outcome variable, the lower risk of stress among women was significant for married (Risk Ratio = 0.63, 95% CI 0.56–0.70) but not for singles (Risk Ratio = 0.95, 95% CI 0.77–1.16). In the model where sleep apnea was the outcome variable, a lower risk of stress among women was significant for married (Risk Ratio = 0.54, 95% CI 0.47–0.61) but not for singles (Risk Ratio = 0.81, 95% CI 0.64–1.03). A lower risk of sleep apnea was also only seen in those with dependent children (Risk Ratio = 0.49, 95% CI 0.42–0.58) but not in those without dependent children (Risk Ratio = 0.71, 95% CI 0.61–0.83).

The risk of stress was also assessed according to overall sleep disorders, insomnia, hypersomnia, and sleep apnea, adjusted for age, sex, marital status, and dependent children (Table 3). The risk of stress was significantly greater for those with a sleep disorder (136%) after adjusting for age, sex, marital status, and dependent children. The risk of stress was also significantly lower for those with insomnia (179%) and for those with sleep apnea (102%) after adjusting for age, sex, marital status, dependent children, and insomnia and hypersomnia. There was not a significant association between the risk of stress and hypersomnia. Interaction terms were assessed but not significant in either model. 

Among contract holders, 1995 (9.5%) had anxiety, 1791 (8.5%) had depression, 422 (2.0%) had ADHD, 123 (0.6%) had bipolar disorder, 84 (0.4%) had OCD, and 23 (0.1%) had schizophrenia. The risk of each of these mental health conditions was significantly associated with stress and sleep disorders (Table 4). In general, stress had a stronger association with the risk of mental health conditions than did sleep disorders, and the combination of stress and sleep disorders had the strongest association. Bipolar disorder and schizophrenia were more strongly associated with stress and sleep disorders than were the other mental health conditions. Insomnia was more strongly associated with anxiety, bipolar disorder, OCD, and schizophrenia than was sleep apnea. 

## 4. Discussion

This study extends previous research by comparing associations between stress and specific types of sleep disorders and by considering whether certain variables modify these associations. It further assesses the association between stress and specific types of sleep disorders with common mental health conditions. The main findings show that insomnia was most strongly associated with stress, followed by sleep apnea. Hypersomnia was not significantly associated with stress. In general, stress more strongly correlates with mental health conditions than do sleep disorders. The strongest associations involve bipolar disorder and schizophrenia. Insomnia was more strongly associated with each of the mental health conditions (except depression and ADHD) than was sleep apnea. 

The risk of stress was associated with sex, marital status, and dependent children. The higher risk of stress among women than men is consistent with other research [29,30,31]. The lower risk of stress in married individuals is consistent with another study in which married people had lower levels of the stress hormone cortisol compared with those who were previously or never married [32]. Spousal support has been shown to attenuate stressful events [33]. Single, separated, widowed, and divorced individuals all reported higher levels of stress, with greater worry about the future and less confidence in their decision-making abilities [34]. Loneliness is also more common in individuals who are not in a current marital relationship. Loneliness increases an individual’s risk of physical and mental health problems and is considered a growing public health concern [35].

Higher stress in those with children is consistent with research on the risk of stress in households with dependents [36]. There is a strong positive association between the number of dependents and the level of stress [36,37]. Having dependent children can cause stress in many forms, including emotional, financial, and psychological. Parental stress has been studied and measured, and the burden of stress has consistently been found to be higher in those with dependents than those without [38,39].

In 2020, the risk of a sleep disorder among adults 18–64 reported in this study was 12.1%. Data from surveys administered to the general population will identify a higher incidence of sleep disorders than found in the current study, where possibly those with more severe conditions resulting in a medical claim were identified as cases. In addition, in contrast to the incidence data reported in this study, cross-sectional surveys typically produce prevalence data, which represents ever having had the problem. Hence, prevalence of sleep disorders will be higher than incidence of sleep disorders. 

The risk of insomnia, hypersomnia, and sleep apnea increased with age, which is consistent with research showing more sleep disturbances occurring in older adults because aging is associated with factors affecting sleep (multimorbidity, polypharmacy, and psychosocial factors) [40,41]. Risk factors for sleep apnea that increase with age include obesity, snoring, hypertension, coronary artery disease, stroke, and diabetes [42].

The risk of insomnia was greater in women, and the risk of sleep apnea was greater in men. Other research has shown a higher risk of insomnia in women than in men [43]. Studies have shown that women experience unique hormonal changes that increase their risk of insomnia [44,45]. Women are more likely to recognize sleep disruptions and report these problems and are also more likely to seek professional help for sleep issues [46]. Research has also shown a higher risk of sleep apnea in men than in women [47]. The reasons why men are more prone to sleep apnea than women may be due to differences in obesity, upper airway anatomy, breathing control, hormones, and aging [48].

Married individuals had a higher risk of sleep apnea, which is likely because the problem is more commonly identified when there is a sleeping partner present [49].

Having dependent children was negatively associated with sleep apnea. This result is unclear. However, risk factors for sleep apnea [42] may be associated with a lower likelihood of having children, possibly contributing to this result.

A significantly increased risk of stress was observed for those with insomnia and sleep apnea but not hypersomnia. The association between stress and insomnia [19,50,51] and sleep apnea is consistent with other studies [10,11,22,25,26]. Failure to find a statistically significant association between stress and hypersomnia is consistent with a lack of other studies reporting an association. However, it seems reasonable that stress can indirectly influence hypersomnia by contributing to other sleep disorders and/or drug or alcohol abuse, which, in turn, is associated with an increased risk of hypersomnia.

A positive association between sleep apnea and stress was significantly greater in singles than in married individuals. This may be because married individuals, especially those whose partner stays in bed with them, are more likely to adhere to overnight treatment [52]. Treatment may help both parties sleep better and lower the potential stress associated with sleep apnea. Further, one study found that divorced individuals tended to have more sleep problems compared with married or single counterparts [53]. Divorce is stressful [54]. Being single because of a divorce stemming in part from sleep apnea may explain some of the higher risk of stress among singles in the current study.

Although many studies have shown that stress is associated with an increased risk of mental health problems (e.g., neurosis, anxiety, depression, bipolar disorder, OCD, and schizophrenia) [55,56,57,58,59,60,61,62,63], the current study looked at the simultaneous association between stress and types of sleep disorders with selected mental health conditions. In general, stress alone had a stronger association with mental health problems than sleep disorders alone, and the combination of stress and sleep disorders had the strongest association. When stress reaches the level that results in sleep difficulties, or sleep difficulties reach the level of producing stress, then other more permanent intrinsic mental health conditions may arise or be aggravated. Hence, it is reasonable that the combination of stress and sleep disorders has the greatest association with the other mental health problems considered.

Bipolar disorder and schizophrenia were more strongly associated with stress and sleep disorders than the other mental health conditions. Schizophrenia was most prone to a combination of stress and sleep disorders. Stress and stress-related sleep disorders are temporary reactions to external factors, which can influence extreme mood swings of emotional highs (mania) and lows (depression), which characterize bipolar disorder. In addition, both bipolar disorder and schizophrenia are associated with psychosis, and severe stress and sleep disorders can cause psychosis [64,65].

The observed positive associations between insomnia and the mental health conditions [66,67,68,69,70] and between sleep apnea and the mental health conditions [71,72,73] are consistent with the literature. In the context of these positive associations, the current study found that insomnia was more strongly associated with anxiety, bipolar disorder, OCD, and schizophrenia than was sleep apnea. This may be because insomnia is more likely to have a bidirectional association with these mental health conditions. Although sleep apnea has been shown to increase the risk of mental health problems, there is no evidence that mental health conditions directly cause sleep apnea. Mental health conditions are not among the established risk factors for sleep apnea (excess weight or obesity, a narrow throat, a round head, hypothyroidism, acromegaly, allergies, deviated septum, medical conditions that congest upper airways, smoking, and alcohol abuse) [74]. However, mental health conditions such as depression may indirectly influence sleep apnea by increasing the risk of obesity, physical inactivity, consuming too much caffeine, smoking, and heavy drinking [23,24].

The study data were based on enrollment and medical claims from enrollees in the DMBA database during 2020, so no selection of subgroups that might have resulted in bias was present. A mental health diagnosis would not lead to losing insurance, so selective under-reporting of stress or other mental health claims for this reason is unlikely. However, it may be that some individuals experiencing stress or other mental health or sleep conditions were not identified based on their filing a claim for treating these problems. Incorrect diagnosis and treatment of mental health and sleep conditions are also possible. The available data did not allow us to identify the level of under- or over-reporting of mental health or sleep conditions. The fact that less serious stress was captured in the claims database could have dampened any association between stress and the mental health and sleep conditions considered.

The data were cross-sectionally evaluated in a single year. Hence, we were not able to evaluate causal directions between stress and sleep disorders. Further, the data did not confirm sleep disorders based on polysomnography records.

## 5. Conclusions

Stress is positively associated with insomnia and sleep apnea, with the association stronger for insomnia. A bidirectional association is more likely between stress and insomnia than between stress and sleep apnea, where stress is more likely to result from sleep apnea rather than cause it (unless indirectly). The positive association between stress and sleep apnea is greater in singles than married people, possibly because married individuals are more likely to adhere to overnight treatment and because singles who have been divorced are more likely to have sleeping disorders, which may have contributed to a stressful divorce. Stress alone has a stronger association with other mental health problems than sleep disorders alone, and the combination of stress and sleep disorders has the strongest association. Bipolar disorder is more strongly associated with stress and sleep disorders than the other mental health conditions. Schizophrenia is more strongly associated with a combination of stress and sleep disorders than the other mental health conditions. Stress and stress-related sleep disorders are temporary reactions to external factors, which characterize bipolar disorder. Both bipolar disorder and schizophrenia are associated with psychosis, and severe stress and sleep disorders can cause psychosis.

The information provided in this study may help health-care workers identify high-risk groups for experiencing stress and sleep disorders. Understanding the association between stress and sleep disorders and potential modifiers of this association can further help identify those at greatest risk. Finally, understanding the individual and combined contributions of stress and sleep disorders to mental health conditions may further inform health-care workers of the importance of treating stress and sleep disorders in order to control mental health conditions. Future research may explore specific factors that can influence both stress and sleep disorders and modify their relationship (e.g., migraine headaches, characteristics of dependent children, such as the age and number of children in the home, social support, and so on).

## Figures and Tables

**Table 1 ijerph-19-07957-t001:** Risk of Stress in Employees According to Selected Demographic Variables, 2020.

	Stress with/without Mental Illness					Stress without Mental Illnesses			
	No.	Stress %	Risk Ratio ^†^	95% LCL ^†^	95% UCL ^†^	Stress %	Risk Ratio ^†^	95% LCL ^†^	95% UCL ^†^
Age									
18–29	2084	2.21	1.00			0.77	1.00		
30–39	4521	2.43	1.21	0.86	1.72	1.26	1.67	0.95	2.93
40–49	5643	2.60	1.30	0.92	1.83	1.22	1.5	0.90	2.77
50–64	8779	1.99	1.00	0.72	1.39	1.06	1.40	0.82	2.40
Sex									
Men	14,419	1.66	1.00			0.87	1.00		
Women	6608	3.60	2.18	1.79	2.65	1.66	2.13	1.61	2.82
Married									
No	4484	3.23	1.00			1.29	1.00		
Yes	16,543	2.01	0.70	0.55	0.89	1.07	0.87	0.60	1.25
Children									
No	7018	2.38	1.00			1.16	1.00		
Yes	14,009	2.22	1.39	1.10	1.76	1.03	1.47	1.04	2.08

Data Source: DMBA. ^†^ Adjusted for the variables in the table. The DMBA insurance database classification for sex as male or female.

**Table 2 ijerph-19-07957-t002:** Risk of Sleep Disorders in Employees According to Selected Demographic Variables, 2020.

	Sleep Disorder %	Risk Ratio ^†^	95% LCL ^†^	95% UCL ^†^	Insomnia %	Risk Ratio ^†^	95% LCL ^†^	95% UCL ^†^
Age								
18–29	2.50	1.00			0.67	1.00		
30–39	6.44	2.46	1.84	3.30	1.02	1.55	0.85	2.83
40–49	11.45	4.36	3.29	5.77	2.25	3.44	1.96	6.02
50–64	17.71	6.74	5.12	8.86	2.96	4.51	2.62	7.75
Sex								
Men	13.49	1.00			2.03	1.00		
Women	9.06	0.69	0.63	0.76	2.35	1.18	0.96	1.46
Married								
No	8.52	1.00			1.98	1.00		
Yes	13.07	1.13	1.00	1.27	2.16	0.96	0.73	1.26
Children								
No	11.29	1.00			2.08	1.00		
Yes	12.51	0.91	0.83	0.99	2.15	1.04	0.82	1.31
	**Hypersomnia** **%**	**Risk Ratio ^†^**	**95% LCL ^†^**	**95% UCL ^†^**	**Sleep Apnea** **%**	**Risk Ratio ^†^**	**95% LCL ^†^**	**95% UCL ^†^**
Age								
18–29	0.53	1.00			1.63	1.00		
30–39	0.53	1.02	0.49	2.10	5.26	2.97	2.08	4.25
40–49	0.99	1.93	0.99	3.77	9.11	5.08	3.59	7.19
50–64	1.38	2.65	1.41	4.96	15.30	8.59	6.12	12.05
Sex								
Men	1.10	1.00			11.73	1.00		
Women	0.80	0.69	0.49	0.97	6.63	0.59	0.53	0.66
Married								
No	0.87	1.00			6.33	1.00		
Yes	1.05	0.97	0.64	1.48	11.15	1.20	1.05	1.38
Children								
No	1.03	1.00			9.18	1.00		
Yes	1.00	0.86	0.61	1.21	10.60	0.90	0.82	0.99

Data source: DMBA. ^†^ Adjusted for the variables in the table.

**Table 3 ijerph-19-07957-t003:** Risk of Stress in Contract Holders According to Sleep Disorders, 2020.

	Stress with/without Mental Illness				Stress without Mental Illnesses			
	Stress %	Risk Ratio ^†^	95% LCL ^†^	95% UCL ^†^	Stress %	Risk Ratio ^†^	95% LCL ^†^	95% UCL ^†^
Sleep Disorder								
No	2.02	1.00			1.08			
Yes	4.02	2.36	1.90	2.94	1.42	1.43	1.00	2.05
Insomnia ^‡^								
No	2.17	1.00			1.09			
Yes	6.94	2.79	1.96	3.98	2.24	1.90	1.02	3.54
Hypersomnia ^‡^								
No	2.26	1.00			1.12			
Yes	3.77	0.84	0.39	1.81	0.94	0.79	0.18	3.43
Sleep Apnea ^‡^								
No	2.10	1.00			1.10			
Yes	3.80	2.02	1.58	2.58	1.32	1.31	0.87	1.97

Data source: DMBA. ^†^ Adjusted for age, sex, marital status, and dependent children. ^‡^ Adjusted for age, sex, marital status, dependent children, and other sleep disorders (insomnia, hypersomnia, and sleep apnea).

**Table 4 ijerph-19-07957-t004:** Risk of Selected Types of Mental Health Conditions in Contract Holders According to Stress and Sleep Status, 2020.

	Anxiety			Depression			ADHD		
	Risk Ratio ^†^	95% LCL ^†^	95% UCL ^†^	Risk Ratio ^†^	95% LCL ^†^	95% UCL ^†^	Risk Ratio ^†^	95% LCL ^†^	95% UCL ^†^
Neither	1.00			1.00			1.00		
Stress	3.66	3.15	4.26	3.62	3.05	4.29	2.45	1.50	4.02
Sleep ^1^	2.55	2.30	2.83	2.91	2.62	3.23	2.68	2.12	3.40
Both	5.84	4.73	7.21	6.98	5.76	8.47	5.93	3.09	11.41
Neither	1.00			1.00			1.00		
Stress	3.35	2.90	3.87	3.29	2.85	3.79	2.55	1.67	3.89
Sleep ^2^	1.95	1.73	2.20	2.40	1.99	2.91	2.43	1.54	3.86
Both	4.66	3.65	5.94	4.63	3.36	6.38	1.83	0.53	6.30
Neither	1.00			1.00			1.00		
Stress	3.29	2.89	3.75	3.26	2.84	3.74	2.55	1.70	3.83
Sleep ^3^	1.18	0.87	1.61	1.08	0.77	1.49	1.83	1.00	3.34
Both	3.04	1.42	6.47	1.74	0.66	4.59	---	---	---
Neither	1.00			1.00			1.00		
Stress	3.35	2.90	3.87	3.66	3.12	4.29	2.20	1.35	3.60
Sleep ^4^	1.95	1.73	2.20	2.55	2.27	2.87	2.20	1.69	2.85
Both	4.66	3.65	5.94	6.59	5.30	8.20	8.34	4.38	15.89
	**Bipolar Disorder**			**OCD**			**Schizophrenia**		
	**Risk Ratio ^†^**	**95% LCL ^†^**	**95% UCL ^†^**	**Risk Ratio ^†^**	**95% LCL ^†^**	**95% UCL ^†^**	**Risk Ratio ^†^**	**95% LCL ^†^**	**95% UCL ^†^**
Neither	1.00			1.00			1.00		
Stress	11.06	6.44	19.01	4.74	2.05	10.93	6.36	1.37	29.52
Sleep ^1^	5.14	3.37	7.84	3.30	1.83	5.97	3.27	1.18	9.10
Both	4.45	1.12	17.63	8.59	2.22	33.33	11.52	1.50	88.57
Neither	1.00			1.00			1.00		
Stress	7.55	4.47	12.75	4.41	2.05	9.50	5.85	1.28	26.85
Sleep ^2^	9.11	5.48	15.15	5.77	2.38	13.97	17.29	5.89	50.71
Both	9.56	2.63	34.68	8.55	1.29	56.66	58.76	6.35	543.32
Neither	1.00			1.00					
Stress	6.12	3.73	10.06	4.55	2.12	9.77	---	---	---
Sleep ^3^	0.94	0.32	2.74	2.44	0.69	8.63	---	---	---
Both	---	---	---	---	---	---	---	---	---
Neither	1.00			1.00			1.00		
Stress	9.79	5.86	16.37	5.77	2.77	12.02	4.86	1.08	21.79
Sleep ^4^	2.92	1.87	4.57	2.51	1.22	5.14	0.47	0.06	3.96
Both	1.00	0.23	4.32	---	---	---	8.36	1.13	61.66

Data source: DMBA. Stress 1: Stress with/without mental illness. Stress 2: Stress without mental illness. --- Insufficient numbers to compute. ADHD: attention deficit hyperactivity disorder; OCD: obsessive-compulsive disorder. ^†^ Adjusted for age, sex, marital status, dependent children, and sleep disorders. ^1^ Any sleep disorder; ^2^ insomnia; ^3^ hypersomnia; ^4^ sleep apnea.

## Data Availability

Data are available upon request to the first author.

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
