# Peer review of "Mental Health Conditions According to Stress and Sleep Disorders"

_ijerph, 2022, doi:10.3390/ijerph19137957_

Round 1
Reviewer 1 Report
The manuscript by Merrill RM evaluated the associations between mental health, stress, and several common types of sleep disorders. Noticeably, whilst the article attempted to investigate the relationships between stress and mental health and sleep disorders, the sampling method failed to distinguish between stress and mental health. Essentially, the population used to study stress (Table 1-3) did not have the individuals with mental disorders removed. Whilst these individuals only took ~10% of the total population investigated, it would be sufficiently high to interfere with the survey results since patients with anxiety/MDD routinely exhibit stressed states. Hence, if possible, the numbers in tables 1-3 should be re-calculated to exclude individuals who had been diagnosed with mental disorders to genuinely reflect the connections between stress and sleeping disorder.
In addition, in the age-grouped tables (tables 1 and 2), it would be more persuasive that, if the data source allows, each age group was shown as categorised by sex (e.g. 18-29, M:1000; F:1084). Since this part argues that men and women have very different stress levels (1.66 vs 3.60), I think we cannot simply take the average of each age group but need to show their statistics (stress %, risk ratio, etc.) by sex.
Apart from the issues I mentioned above, the manuscript is generally well written with great mastering of language. I look forward to seeing the revised version of the manuscript in the next submission.
Author Response
Tables 1 and 3 now include analysis for stress only. Also, the point on sex by age is good, but we did not find an interaction involving sex and age for the analysis conducted for both tables. Please see modified text on lines 131-137 and 152-159.
Reviewer 2 Report
The paper presented to me for review "Mental Health Conditions According to Stress and Sleep Disorders" is an original work assessed associations between stress and selected types of sleep disorders, the modifying effect of certain demographic variables on these associations, and the risk of other selected mental health conditions by combinations of stress and sleep disorders. Analyses were based on 21,027 employees aged 18-64 years.
The authors showed that bipolar disorder and schizophrenia were more strongly associated with stress and sleep disorders than were the other mental health conditions. Insomnia was more strongly associated with anxiety, bipolar disorder, OCD, and schizophrenia than was sleep apnea.
The paper is clearly written, is well planned, and most importantly brings new knowledge to this topic. I have one comment - were migraine sufferers included in the analysis? It is the most common neurological disease in the World, significantly affecting daily functioning and often coexisting with mental illness. The association of migraine with sleep disorders has been shown to be significant: https://pubmed.ncbi.nlm.nih.gov/34073933/
Author Response
At the end of the conclusion the following was added: “Future research may explore specific factors that can influence both stress and sleep disorders and modify their relationship (e.g., migraine headaches, characteristics of dependent children like age and number of children in the home, social support, and so on).”
Reviewer 3 Report
The authors perform a regression analysis of insurance data to identify an association between ICD-10 for stress, sleep disorders and mental illness.
Following suggestions should be considered to improve the manuscript:
Before we analyze individual sections of the study, the premise of why this study was even conducted should be addressed.
Methods:
C1. Can the authors define stress? Most other conditions measured and reported in this study can be readily defined.
C2. Please comment on how this study adds to the body of scientific literature.
Abstract:
C3. The abstract should better specify what the objectives of your study are and how you intend on completing these. Please consider editing the sentence, “This study assessed associations…stress and sleep disorders.” Consider using the objectives, methods, main results and conclusion format.
Introduction:
C4. The introduction will benefit from better focus.
Statistical analysis:
C5. Describe statistical analysis separately. How did the authors arrive at the decision of using age, sex, marital status, and dependent children as covariates for adjustment in their regression models? Why not adjust for other sleep disorders while analysis an association between say stress and insomnia. Similar argument can be made for the relation between stress and mental illness. Explain this in your statistical analysis.
Results:
C6. Please define men and women in the results. Readers may object to reporting just men.
C7. Table 1 ideally should be demographic characteristics only. Any subsequent analysis should be defined in Table 2 onwards.
Discussion:
C8. The 1st paragraph of your Discussion should enlist your main findings and explain how these were significant vs. known literature on the subject.
C9. Prevalence of sleep disorders was lower than in the general population. This should be addressed in discussion.
C10. Lack of sleep study data should be acknowledged in limitations.
Author Response
The authors perform a regression analysis of insurance data to identify an association between ICD-10 for stress, sleep disorders and mental illness.
Following suggestions should be considered to improve the manuscript:
Before we analyze individual sections of the study, the premise of why this study was even conducted should be addressed.
Response: The final paragraph of the Introduction was replaced with the following: “This study was conducted to better understand the association between stress and sleep disorders (insomnia, hypersomnia, and sleep apnea), identify potential modifying effects, and determine the relative influence of stress alone, sleep disorders alone, and stress and sleep disorders combined on selected types of mental health illness.”
Methods:
C1. Can the authors define stress? Most other conditions measured and reported in this study can be readily defined.
Response: In paragraph 3 of the Methods, the following sentence was added: “Stress (F43) consists of reaction to severe stress and adjustment disorder. It is applicable to acute crisis reaction, acute reaction to stress, combat and operational stress reaction, combat fatigue, crisis state, and psychic shock.”
C2. Please comment on how this study adds to the body of scientific literature.
Response: The first paragraph of the Discussion now says: “This study extends previous research by comparing associations between stress and specific types of sleep disorders and by considering whether certain variables modify these associations. It further assesses the association between stress and specific types of sleep disorders with common mental health conditions. The main findings show that insomnia is most strongly associated with stress, followed by sleep apnea. Hypersomnia is not significantly associated with stress. In general, stress more strongly correlates with mental health conditions than do sleep disorders. The strongest associations involve bipolar disorder and schizophrenia. Insomnia more strongly associates with each of the mental health conditions (except depression and ADHD) than is sleep apnea.”
Abstract:
C3. The abstract should better specify what the objectives of your study are and how you intend on completing these. Please consider editing the sentence, “This study assessed associations…stress and sleep disorders.” Consider using the objectives, methods, main results and conclusion format.
Response: The first sentence of the Abstract now says: “The purpose of this study was to compare associations between stress and sleep disorders (insomnia, hypersomnia, and sleep apnea), identify potential modifying effects, and compare associations between stress and types of sleep disorders with selected mental health conditions.”
It does not appear that this journal uses a structured abstract in the prescribed format, but rather a single paragraph.
Introduction:
C4. The introduction will benefit from better focus.
Response: The final paragraph of the Introduction was rewritten to provide better focus.
Statistical analysis:
C5. Describe statistical analysis separately. How did the authors arrive at the decision of using age, sex, marital status, and dependent children as covariates for adjustment in their regression models? Why not adjust for other sleep disorders while analysis an association between say stress and insomnia. Similar argument can be made for the relation between stress and mental illness. Explain this in your statistical analysis.
Response: Sleep disorders is now adjusted for in the analyses. This sentence was added in the statistical techniques section: “Risk ratios measuring the association between stress and each specific type of sleep disorder further adjusted for the other types of sleep disorders.”
The Methods section was divided into three subsections: Study Population, Data Collection, and Statistical Techniques. The following was added after the first sentence in the “Statistical Techniques” section.
Results:
C6. Please define men and women in the results. Readers may object to reporting just men.
Response: The following was added to the footnote in Table 1: “The DMBA insurance databased used the classification for sex as male or female.”
C7. Table 1 ideally should be demographic characteristics only. Any subsequent analysis should be defined in Table 2 onwards.
Response: Because having children in the home can impact both parental stress (see Table 1) and sleep quality (see Table 2), it is hoped that this variable can remain in the first table.
Discussion:
C8. The 1st paragraph of your Discussion should enlist your main findings and explain how these were significant vs. known literature on the subject.
The first paragraph of the Discussion now says: “This study extends previous research by comparing associations between stress and specific types of sleep disorders and by considering whether certain variables modify these associations. It further assesses the association between stress and specific types of sleep disorders with common mental health conditions. The main findings show that insomnia is most strongly associated with stress, followed by sleep apnea. Hypersomnia is not significantly associated with stress. In general, stress more strongly correlates with mental health conditions than do sleep disorders. The strongest associations involve bipolar disorder and schizophrenia. Insomnia more strongly associates with each of the mental health conditions (except depression and ADHD) than is sleep apnea.”
C9. Prevalence of sleep disorders was lower than in the general population. This should be addressed in discussion.
Response: The following was added after paragraph 3 of the Discussion: “In 2020, the risk of a sleep disorder among adults 18-64 reported in this study was 12.1%. Survey data administered to the general population will identify higher incidence of sleep disorders than found in the current study, where only those with more severe conditions involving a medical claim were identified as cases. In addition, survey data typically report prevalence of a sleep disorder, which represents ever having had the problem. Hence, prevalence estimates of sleep disorders will be higher.”
C10. Lack of sleep study data should be acknowledged in limitations.
Response: The following was added to the limitations: “Further, the data did not confirm sleep disorders based on polysomnography records.”
Reviewer 4 Report
Dear Author, thank you for this paper. This paper presents the associations between stress and sleep disorders and according mental health conditions. It was a pleasure to read this manuscript. In order to improve this paper, a few comments for the author:
-line 105: dependent children: how are these children determined as dependent? where is this stated?
-results: stress: please define more clearly how this parameter was questionned. Was it assessed by physicians, by the insurance company? An how was it assessed? If this is not clearly mentioned, please indiciate why it is not clear how and by whom it was assessed and include thisin your discussion.
-same comment is reaise for insomnia, hypersomnia and sleep apnea
-young adults: was it questionned whether they were studying or working? was it questionned whether they were living with parents or alone? And if not, motivate why not. Or include this in the discussion
-adults: was it questionned whether they were living togheter (eg with friends or siblings) or living alone? And if not, motivate why not. Or include this in the discussion.
-where the ages of the children of these adults reported? Peolpe with infants can experience different stress levels than parents of toddlres or other dependent children.
-line 172: please also refer to references stating that sleep apnea was greater in men
-line 184: please split up the references that are refering to insomnia and sleep apnea
-line 216: please split up the references that are refering to insomnia and sleep apnea
-discussion: the clinical relevance of the findings in this paper are missing. Please complement this discussion with the clinical relevance and implications
-what are the suggestions for future research? Please include these in your discussion and conclusion.
-conclusion: lines 253-257 are redundant. Please remove these lines, or include them in your results
Author Response
Dear Author, thank you for this paper. This paper presents the associations between stress and sleep disorders and according mental health conditions. It was a pleasure to read this manuscript. In order to improve this paper, a few comments for the author:
-line 105: dependent children: how are these children determined as dependent? where is this stated?
Response: The following was added (lines 105-107): “A dependent child was defined as an individual residing in the contract holder’s household who can legally be claimed as a dependent on the federal income tax.”
-results: stress: please define more clearly how this parameter was questionned. Was it assessed by physicians, by the insurance company? An how was it assessed? If this is not clearly mentioned, please indiciate why it is not clear how and by whom it was assessed and include this in your discussion.
Response: On line 96, the following was added: “Licensed health-care providers submitted medical claims to the insurance company.”
On lines 97-99, the following was added: “Stress (F43) consists of reaction to severe stress and adjustment disorder. It is applicable to acute crisis reaction, acute reaction to stress, combat and operational stress reaction, combat fatigue, crisis state, and psychic shock.”
-same comment is reaise for insomnia, hypersomnia and sleep apnea
Response: Insomnia, hypersomnia, and sleep apnea are defined on lines 43-45. Also, the new sentence on line 96 should address your comment.
-young adults: was it questionned whether they were studying or working? was it questionned whether they were living with parents or alone? And if not, motivate why not. Or include this in the discussion
Response: The study focuses on contract holders; employees. Also, please see our added definition of dependent children on lines 105-107.
-adults: was it questionned whether they were living togheter (eg with friends or siblings) or living alone? And if not, motivate why not. Or include this in the discussion.
Response: While the contract holders in which this study focuses were assessed according to whether they were married and if they had children in the home, data were not available through the insurance database on whether they were living with friends or siblings. The following was added at the end of the conclusion: “Future research may explore specific factors that can influence both stress and sleep disorders and modify their relationship (e.g., migraine headaches, characteristics of dependent children like age and number of children in the home, social support, and so on).”
-where the ages of the children of these adults reported? Peolpe with infants can experience different stress levels than parents of toddlres or other dependent children.
Response: Note the sentence that was included at the end of the Conclusion on future research.
-line 172: please also refer to references stating that sleep apnea was greater in men
Response: Reference 43 was added, which study also says that women are more likely to experience insomnia than men. Reference 47 was added, which study also says that men are more likely to experience sleep apnea than women.
-line 184: please split up the references that are refering to insomnia and sleep apnea
Response: The references are now split. In addition, two references were added that show an association between stress and insomnia.
-line 216: please split up the references that are refering to insomnia and sleep apnea
Response: The references are now split.
-discussion: the clinical relevance of the findings in this paper are missing. Please complement this discussion with the clinical relevance and implications
Response: The following paragraph was added at the end of the Conclusion: “The information provided in this study may help health care workers identify high-risk groups for experiencing stress and sleep disorders. Understanding the association between stress and sleep disorders and potential modifiers of this association can further help identify those at greatest risk. Finally, understanding the individual and combined contributions of stress and sleep disorders to mental health conditions may further inform health care workers of the importance of treating stress and sleep disorders in order to control mental health conditions. Future research may explore specific factors that can influence both stress and sleep disorders and modify their relationship (e.g., migraine headaches, characteristics of dependent children like age and number of children in the home, social support, and so on).”
-what are the suggestions for future research? Please include these in your discussion and conclusion.
Response: see previous response.
-conclusion: lines 253-257 are redundant. Please remove these lines, or include them in your results
Response: The lines were modified to better match the findings reported in the Results.